# Product- and Hydro-Validation of Satellite-Based Precipitation Data Sets for a Poorly Gauged Snow-Fed Basin in Turkey

Gökçen Uysal 

Department of Civil Engineering, Eskişehir Technical University, Eskişehir 26555, Turkey;
gokcenuysal@eskisehir.edu.tr

**Abstract:** Satellite-based Precipitation (SBP) products are receiving growing attention, and their utilization in hydrological applications is essential for better water resource management. However, their assessment is still lacking for data-sparse mountainous regions. This study reveals the performances of four available PERSIANN family products of low resolution near real-time (PERSIANN), low resolution bias-corrected (PERSIANN-CDR), and high resolution real-time (PERSIANN-CCS and PERSIANN-PDIR-Now). The study aims to apply Product-Validation Experiments (PVEs) and Hydro-Validation Experiments (HVEs) in a mountainous test catchment of the upper Euphrates Basin. The PVEs are conducted on different temporal scales (annual, monthly, and daily) within four seasonal time periods from 2003 to 2015. HVEs are accomplished via a multi-layer perceptron (MLP)-based rainfall-runoff model. The Gauge-based Precipitation (GBP) and SBP are trained and tested to simulate daily streamflows for the periods of 2003–2008 and 2009–2011 water years, respectively. PVEs indicate that PERSIANN-PDIR-Now comprises the least mean annual bias, and PERSIANN-CDR gives the highest monthly correlation with the GBP data. According to daily HVEs, MLP provides a compromising alternative for biased data sets; all SBP models show reasonably high Nash–Sutcliffe Efficiency for the training (above 0.80) and testing (0.62) periods, while the PERSIANN-CDR-based MLP (0.88 and 0.79) gives the highest performance.

**Keywords:** satellite precipitation; neural network model; rainfall-runoff application; snowmelt; water resources; upper Euphrates basin

## 1. Introduction

Decision support systems are vital, especially in the snow-fed mountainous basins where large reservoirs located downstream are generally used for hydropower, irrigation, etc. for effective management and preparedness. In these basins, snowmelt contributes greatly to the runoff; thus, the main component of decision support systems snow modeling and forecasting are crucial tasks that require a proper precipitation input for hydrological monitoring and hydrological forecasting [1–3]. The quality of the precipitation data is associated with the proper installation of the observation network. Even though a denser station network is recommended by the World Meteorological Organization (WMO), such as one station per 250 km$^2$ [4], installation and maintenance are still challenging missions for the harsh topography (steep topography, power supply conditions, etc.). In addition, the measurement network is a costly item for many developing countries. Radar systems can provide higher temporal and spatial resolution data, yet they are expensive to install [5] and suffer from various discrepancies (blockage by mountain obstructions, beam overshoot, anomalous propagation, etc.) in complex topography [6]. For a couple of decades, by means of advances in remote sensing technology, global high-resolution Satellite-based Precipitation (SBP) products have been developed and have become operational. These data sets can be based on direct satellite data, reanalysis products, or merged precipitation products.

Although these SBP data sets are readily available, they still suffer from positive or negative biases [7–13]. The accuracy of these products is related to the location of

interest since validation and bias correction are mainly utilized for data-rich regions. For example, the accuracy of PERSIANN [14] shows a higher performance in the continental United States compared to Turkey [11,13]. The validation of precipitation is more complex compared to temperature, so there are different performance assessment approaches. For example, the performances of various SBP data sets (GPM IMERGv05, TMPA 3B42V7, ERA-Interim, and ERA5) are assessed across Turkey by considering different slopes and wetness classes [15]. The performances of eight high-resolution gridded precipitation products (CMORPHv1-CRT, CRU TSv.4.05, ERA5, GSMaP_NRT, IMERG V06B-Final, MSWEPv2.0, PERSIANN-CDR, and TRMM 3B42v7) are compared for annual and monthly temporal scales, especially over mountainous areas such as the Heihe River basin, China [16]. Due to the complexity of the precipitation phenomena, the validation in a region can differ from another region. For example, the performance of the PERSIANN is lower due to the warmer cloud formation over the western contiguous United States (CONUS) [17]. Another issue might be attributed to the seasonal performances of the SBP such that different satellite precipitation estimates and the error sources are systematically different for different seasons [11,15,18]. Therefore, their performance assessment in the selected different pilot basins still provides valuable insight for the operators and managers.

Since estimation biases can limit their reliability applications, the restricted studies also consider the application of hydrological monitoring and modeling together with SBP utilization. For example, five SBP data (TMPA-RT, TMPA-V6, CMORPH, PERSIANN, and adjusted PERSIANN) are compared as forcing data for streamflow simulations at 6-hourly and monthly time scales over the mid-size Illinois River basin, the United States [7]. In another study, four different SBPs (3B42, CMORPH, 3B42RT, and PERSIANN) are evaluated with gauge observations over the Tibetan Plateau (limited within China) in streamflow simulations using the variable infiltration capacity (VIC) hydrological model [18]. One study tests the Soil and Water Assessment Tool (SWAT) to mimic the streamflow in a subbasin of the Tocantins river basin using SBP of TRMM, TMPA, and IMERG and rain gauges in a basin of the Cerrado biome, Brazil [19]. The three-stage dynamic scheme is proposed and compared to individual precipitation datasets (IMERG Final, TMPA 3B42V7, and PERSIANN-CDR), and the improved data sets are later employed in hydrological models for some catchments over China by improving the Kling-Gupta Efficiency (KGE) of simulated streamflow by 12–35% [20].

Multi-sourcing merging data set SBP sometimes can provide similar or better performance in hydro-validation studies in comparison to streamflow gauging measurement. Luo et al. [21] demonstrate that TRMM and CHIRPS-based daily and monthly hydrological simulations via the SWAT model produced improved results against gauge-based and inverse distance weighted data in terms of Nash–Sutcliffe Efficiency (NSE) for four different basins in the Lower Lancang-Mekong River Basin, China. Ahady et al. [22] apply GR2M monthly hydrological modeling, and CHIRPS-2.0 performs better than gauge-based and GPM-IMERG-V6 precipitation. Another study [13] shows that the daily HBV model shows that MSWEP V2.8 gives similar NSE, and CHIRPS V2.0 gives slightly higher KGE against gauge-based modeling for the entire period. Zubieta et al. [23] analyze TMPA, CMORPH, PERSIANN, and rain gauge data for the poorly gauged Western Amazon basin of Peru and Ecuador. They developed the daily MGB-IPH hydrological model at 18 streamflow gauging stations where the drainage basin area of the studied streamflow gauges varied (4613–878,306 km$^2$) and there is a large range in the annual flow (599–35,569 m$^3$/s). In their assessment, both gauge-based and TMPA SBP show similar model performances in hydrological simulations and are better than those using CMORPH and PERSIANN. Le et al. [5] emphasize the outperformance of SBP-based monthly hydrological simulations against gauge-based simulations in large Vietnam basins using the SWAT. This is most likely due to the low-density stations and the poor quality of rainfall data.

SBP data are continually enhanced over time by using new algorithms, different data sets (microwave, radar, etc.), and ground-based measurements. SBP algorithms use a set of diverse data from various sensors onboard geosynchronous-Earth-orbiting satellites (GEOs)

and low-Earth-orbiting satellites (LEOs). Satellite observations do not offer direct readings of precipitation amount; instead, an indirect association is employed to link the probability and intensity of rainfall to data collected by one or more sensors. The PERSIANN (Precipitation Estimation from Remotely Sensed Information Using Artificial Neural Networks) system [14,24] computes an approximate rainfall rate at each $0.25° \times 0.25°$ pixel of the infrared brightness temperature photo provided by GEO satellites using neural network function classification/approximation methods by the Center for Hydrometeorology and Remote Sensing (CHRS) at the University of California, Irvine (UCI). LEO passive microwave information data (Instantaneous rainfall estimates from TRMM, NOAA, and DMSP satellites, etc.) are used to train the model (parameter estimation).

The Artificial Neural Network (ANN) model has been a well-known concept in hydrology for a couple of decades [25–28]. It is capable of mapping nonlinear relationship and is applied in many regions all over the world [29–33]. One of the architectures of ANN, the Multi-layer Perceptron Model (MLP), is also selected to conduct the rainfall–runoff relationship in this study. A recent study [13] compared 13 products (CPCv1, MSWEPv2.8, ERA5, CHIRPSv2.0, CHIRPv2.0, IMERGHHFv06, IMERGHHEv06, IMERGHHLv06, TMPA-3B42v7, TMPA-3B42RTv7, PERSIANN-CDR, PERSIANN-CCS, and PERSIANN) over a mountainous Karasu basin in Turkey. They claim that all SBP data perform better in producing streamflow when the model is calibrated by each SBP separately. Similar to this concept, SBP data and Gauge-based Precipitation (GBP) are trained and tested separately to gain better performances. This study is also conducted in another of the tributaries of the Euphrates River Basin (Murat), but the selection of a least studied and a different application region due to its remote location with data gap, the application of the longer and more historical time period, the selection of a different hydrological model structure, and the addition of new SBP data (PERSIANN-PDIR-Now) can be considered as the new contributions and the novel parts of my study on top of Hafizi and Sorman [13].

Although hydro-validation using various rainfall-runoff models has gained popularity together with product-validation in the literature, most of the studies are conducted with mainly process-based hydrological models in semi-humid/humid climates [34–36], rain-dominated basins [37–41], or snow-dominated regions without accounting for snowmelt module [42,43]. There are several product-validation studies in snow-dominated regions [44–46], but the application of both product- and hydro-validation in snow-dominated regions is still limited [11–13,21,47]. Additionally, there is only very little assessment for newly developed PERSIANN-PDIR-Now SBP [48–50]. Therefore, this study differs from others in terms of both product- and hydro-validation by integrating SBP with the ANN-based model to generate daily streamflows for remotely located, snow-dominated, and poorly gauged regions. The study reveals the performances of PERSIANN family products (PERSIANN, PERSIANN-CDR, PERSIANN-CCS, and PERSIANN-PDIR-Now) in the upper mountainous region of the Euphrates River Basin (Tutak Basin). In order to establish a rainfall–runoff relationship, a data-driven ANN model is employed training the best architecture with observed runoff. The innovative parts of the study are to account for both ground-based comparison in terms of the annual, monthly, and daily scale for four split seasons considering snowmelt processes and the implementation of ANN-based models separately to simulate daily streamflows obtained by SBP data and compared with a ground-based model.

## 2. Materials and Methods

The study consists of two main steps, as shown in Figure 1. The first is based on the validation of the SBP data sets in comparison with GBP, and the latter utilizes SBP in hydrological modeling to assess the hydro-validation. The analyses are also assessed in various seasons with regard to the local seasons of the study area.

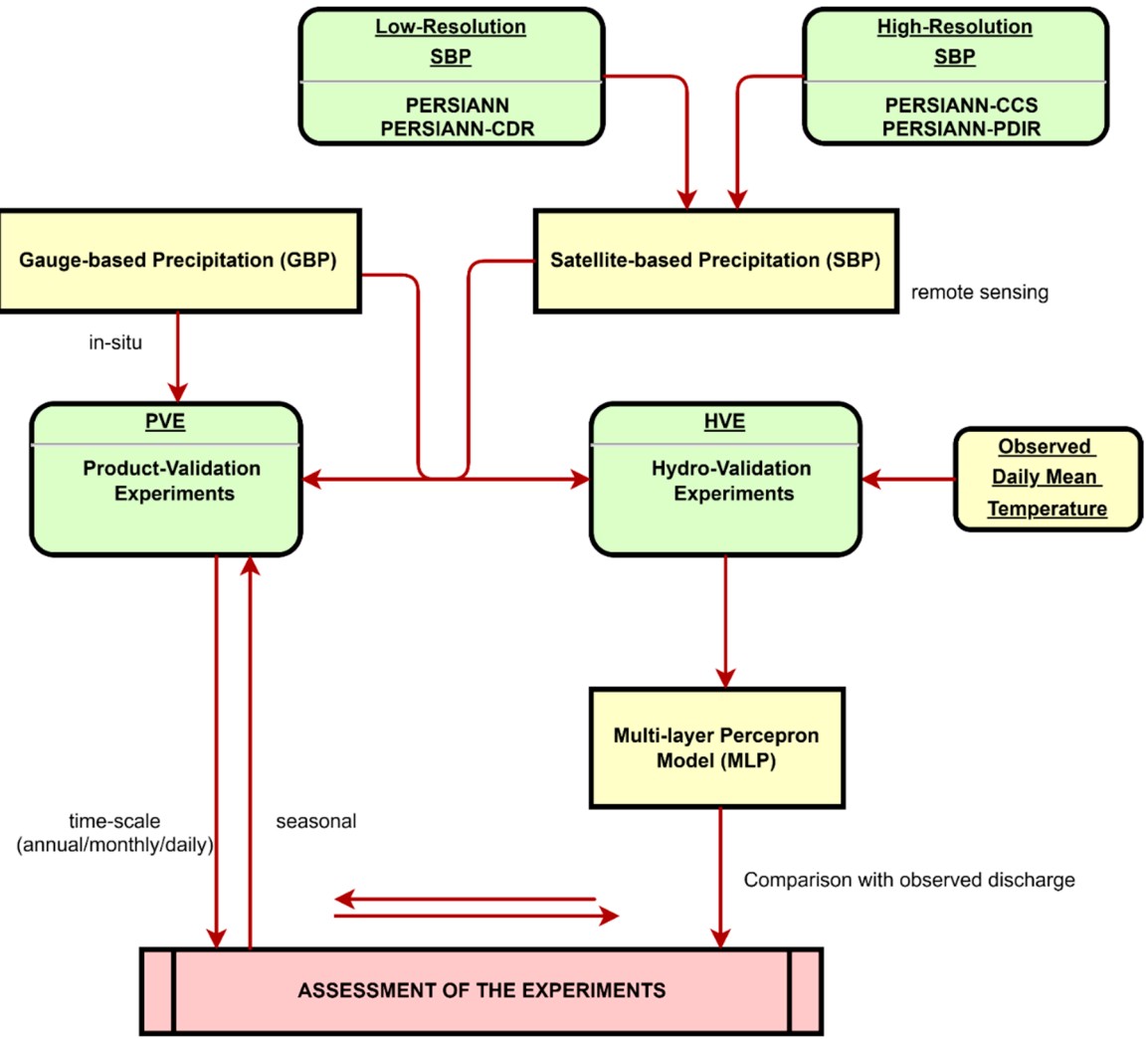

**Figure 1.** The flowchart of the study.

*Product-Validation Experiment (PVE):* This experiment assesses the accuracy of the SBP data sets versus GBP. GBP is provided from long historical data sets of the meteorological station located in the basin. The experiments are conducted on different time scales, i.e., yearly, monthly, and daily.

*Hydro-Validation Experiment (HVE):* This experiment assesses the effects of the SBP on streamflow. Since the snow accumulation and melting processes are dominant in the mountainous pilot basin, the assessment is expected to reveal the performance of the SBP data sets on the snowmelt runoff as well. The hydrological modeling is accomplished with a data-driven model and applied on a daily basis. The inputs to the hydrological do not include the lagged values of the previously observed streamflow data. The models only include meteorological inputs as forcing to the model inputs. To that end, independent daily MLP model-based configurations are conducted. This means each experiment is trained and tested with its own data; this also eliminates the requirement of the bias correction for the independent input sets. Thus, the model parametrization is adjusted with the original data. For the sake of consistency among the model sets and to reveal the effect of the precipitation, the same temperature data is used in the model configurations.

## 2.1. Satellite-Based Precipitation Products

The PERSIANN family products are gaining popularity across the world. Several studies have been conducted using the PERSIANN family SBP with information about the accuracy of estimates [11,13,14,48,50,51]. Salehi et al. [48] emphasize that each of the

PERSIANN family datasets has a special skill. There are five products available in total at present and the four types of SBP products are utilized in this study as PERSIANN, PERSIANN-Cloud Classification System (PERSIANN-CCS), PERSIANN Climate Data Record (PERSIANN-CDR), and PERSIANN Dynamic Infrared Rain Rate (PERSIANN-PDIR-Now). The remaining PERSIANN-CCS-CDR is recommended for heavy precipitation studies [36,52]. Therefore, PERSIANN-CCS-CDR is considered out of the scope and not included in the study. The basic attributes of the PERSIANN family products are given in Table 1. The study is carried out on a 1-day scale, considering the temporal resolution of all products. The data sets can mainly be categorized in terms of their resolution: low spatial resolution SBP (PERSIANN and PERSIANN-CDR) and high spatial resolution SBP (PESIANN-CCS and PERSIANN-PDIR-Now). PERSIANN-CDR is also a bias-corrected one among all SBP data sets.

**Table 1.** Algorithm and basic attributes of the SBP products.

| Feature | Low-Resolution SBP | | High-Resolution SBP | |
|---|---|---|---|---|
| | **PERSIANN** | **PERSIANN-CDR** | **PERSIANN-CCS** | **PERSIANN-PDIR-Now** |
| Time delay | 2-day delay | ~3-month delay | Near real-time with 1 h delay | Near real-time with 1 h delay |
| HTTP Download (full globe) | hourly, 3-hourly, 6-hourly, daily, monthly, yearly | daily, monthly, yearly | hourly, 3-hourly, 6-hourly, daily, monthly, yearly | hourly, 3-hourly, 6-hourly, daily, monthly, yearly |
| Bias correction | No | Yes | No | No |
| Data used for bias correction | - | GPCP monthly precipitation data $(2.5° \times 2.5°)$ | - | - |
| Availability period | March 2000–Present | January 1983–Present | January 2003–Present | March 2000–Present |
| Coverage | 60°S-60° N (quasi global) | 60° S-60° N (quasi global) | 60° S-60° N (quasi global) | 60° S-60° N (quasi global) |
| Temporal resolution | Hourly | Daily | Hourly | Hourly |
| Spatial resolution | $0.25° \times 0.25°$ | $0.25° \times 0.25°$ | $0.04° \times 0.04°$ | $0.04° \times 0.04°$ |

The PERSIANN-CCS system categorizes cloud-patch properties based on cloud height, areal extent, and textural variability derived from satellite data using the variable threshold cloud segmentation algorithm [17]. Therefore, it is claimed by the model developers that PERSIANN-CCS detects the spatial distribution of precipitation better than the traditional PERSIANN. On the other hand, PERSIANN-CDR [53] is a pre-bias-corrected product with global precipitation datasets of the Global Precipitation Climatology Project (GPCP). PERSIANN-PDIR-Now is an improved version of PERSIANN-CCS and a real-time global high-resolution product of the high frequency sampled longwave infrared (IR) imagery [54]. PERSIANN family SBP products are freely assessed from an easily accessible public repository [55]. With SBP over data-sparse regions and complex terrain, gauge adjustment could result in no improvement compared to satellite-only rainfall products [56]. Considering this and accounting for the MLP model in the study, which can adjust the parameter sets based on the training step, no gauge adjustment is applied.

*2.2. Study Area and Hydro-Meteorological Data Set*

The Euphrates River Basin (127,300 km$^2$) has major branches as Karasu, Murat, Peri, and Munzur into Keban Dam. Forecasting the amount and timing of discharge at the headwaters of the Euphrates River in Eastern Anatolia of Turkey has always had great importance for the operation of downstream reservoirs. This study is conducted in the Upper Murat (Tutak) Basin placed in the upper part of the Murat River Basin (a major branch

of the Euphrates River, 40,000 km² in total). Murat River brings a higher percentage of water to the Euphrates River. The study basin is within the longitudes 42°00′ E to 44°00′ E, latitudes 39°00′ N to 40°00′ N, and the basin is controlled by the stream gauging station (station ID: E21A022) at Tutak location with a drainage area of 5910 km², and its elevation ranges in altitude from 1559 to 3508 m. The main land cover types are pasture (32.6%), agriculture (36.1%), bareland (30.8%), and the others (urban, forest, lakes, etc., 0.6%). The location and elevation ranges of Tutak Basin along with the observation network are given in Figure 2.

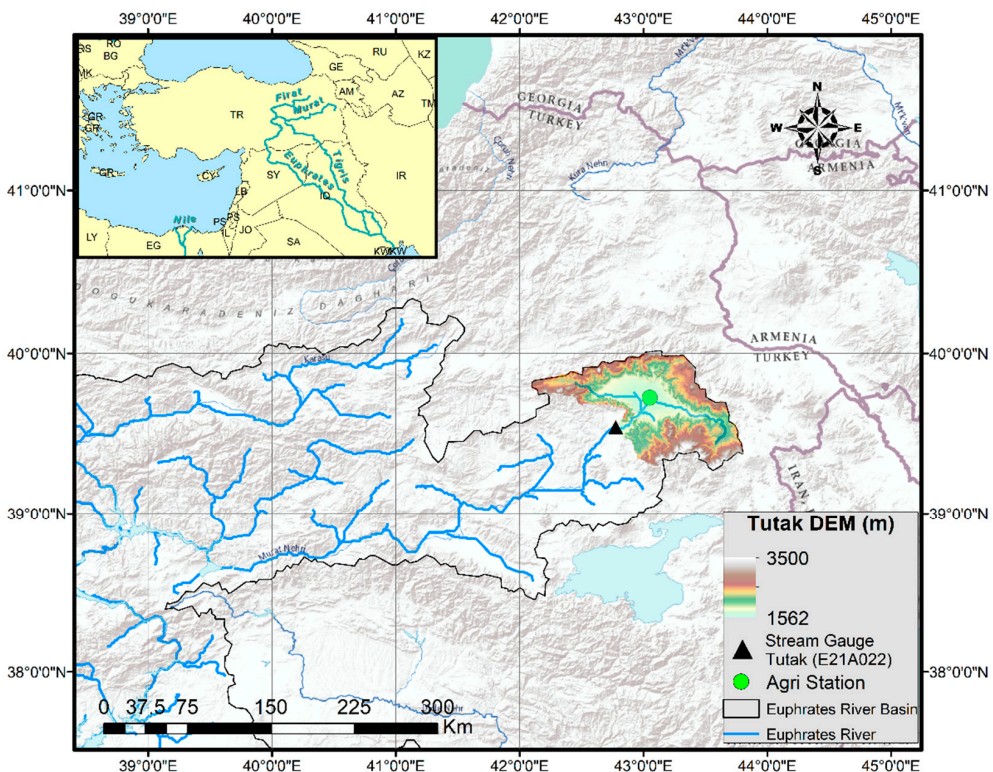

**Figure 2.** Location, digital elevation model, and the observation network of the Upper Murat (Tutak) Basin.

The catchment has a limited observation network because of the steep terrain, harsh weather condition, and high altitude. Indeed, this is the one main motivation to check the suitability of the SBP in the selected region. As illustrated in Figure 2, the watershed has only one meteorological station (Agri at an altitude of 1632 m) that records daily precipitation (P) and temperature (T) within the basin boundary. The study assumes that the meteorological station represents the basin averages of the meteorological variables.

The variation of daily average precipitation data, temperature data and streamflow observations is demonstrated in Figure 3. The meteorological data is provided by the Turkish State Meteorological Service (Meteoroloji İşleri Genel Müdürlüğü, MGM), and the streamflow data is provided by the State Hydraulic Works (Devlet Su İşleri, DSI). In view of a mutual date range, the start date for the analyses (PVEs and HVEs) is January 2003 due to the availability of PERSIANN-CCS data. The end date for the PVEs is May 2015 due to the data availability from the Agri station. Average annual precipitation and temperature values are 488 mm and 6.7 °C, respectively, for the available long-term records (January 2003–May 2015). On the other hand, the streamflow data is missing between 2012 and 2015; therefore, HVEs are conducted from 2003 to 2011. The mean streamflow from 2003 to 2011 equals 59.4 m³/s, while it reaches a mean of 152.9 m³/s during the melting season.

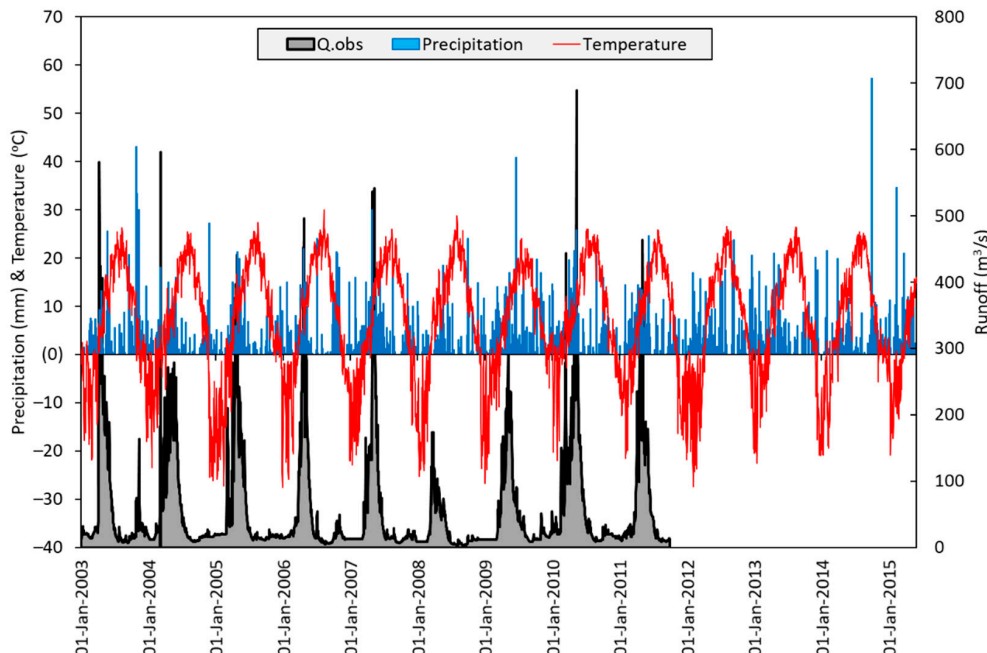

**Figure 3.** Daily precipitation and average air temperature records of Agri station and streamflow observation from Tutak stream gauge.

The assessments of the experiments are conducted for four different seasons, and the details are given in Table 2: "water year" (one annual cycle, the water year concept in the Northern Hemisphere, WY), "snow accumulation season", "snow melting season", and "summer season" or so-called "no-snow season". The mean flow values indicate the importance of the melting season for the area of interest.

**Table 2.** Seasons for the analyses.

| Season | Start | End | Local Season | Number of days | Mean Flow (m³/s) |
|---|---|---|---|---|---|
| Water Year | 1 October | 30 September | One full water year | 365 | 59.4 |
| Snow Accumulation | 1 October | 28 February | Late fall and full winter | 151 | 20.3 |
| Snow Melting | 1 March | 31 May | Full spring | 92 | 152.9 |
| No-Snow | 1 June | 30 September | Full summer and early fall | 122 | 34.0 |

Figure 4 demonstrates the visual comparison of the SBP cells across the study area for one of the randomly selected dates. In this study, all SBP data are analyzed according to their original spatial resolutions. The variation of the precipitation distribution within the SBP is noticeable. There are 10 cells within the basin boundary for PERSIANN and PERSIANN-CDR and 391 cells for PERSIANN-CCS and PERSIANN-PDIR-Now. The basin average values within the catchment are extracted for each day, and daily time series are obtained for 2003–2015.

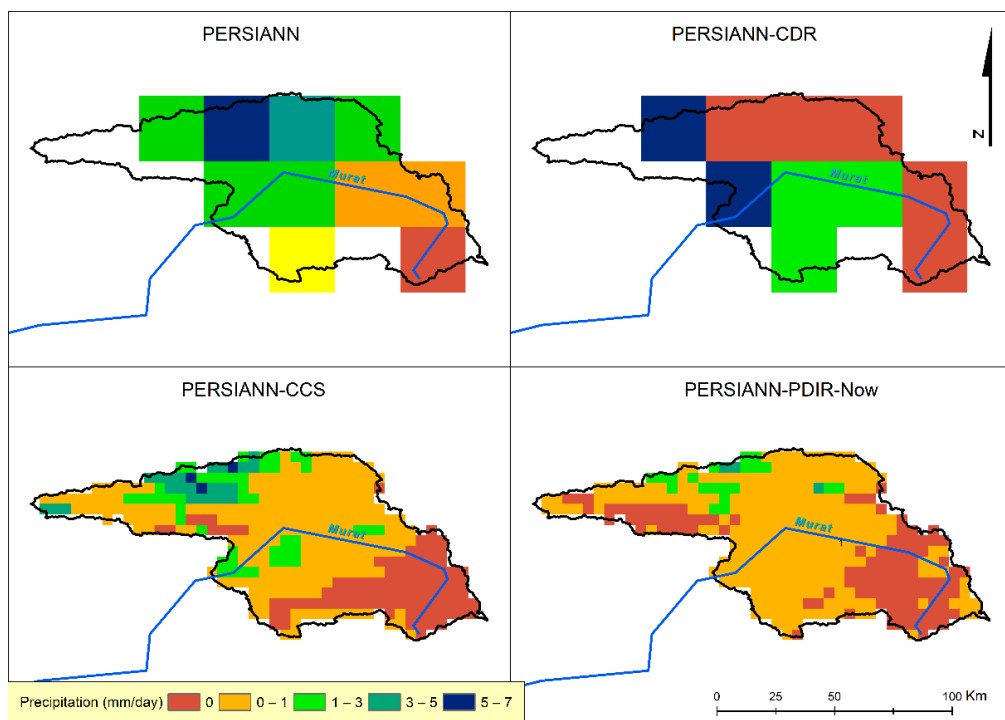

**Figure 4.** SBP distribution across the study area for 5 January 2015.

### 2.3. Multi-Layer Perceptron Model

A reliable rainfall-runoff model is essential to test the hydro-validation of the SBP products. In this study, a data-driven model is selected and applied to utilize SBP data sets due to two main reasons. First of all, physical and conceptual models (especially considering the vertical variation of the topography in the mountainous catchments) require a detailed and rich data network. However, this may increase the demand for SBP data sets in each elevational band. Because this is not suitable with some SBP data sets due to geographical resolution, a data-driven strategy is chosen as an alternative due to its ease of implementation [11,32].

On the other hand, considering the resolution, variability of the topography and some other factors, a bias correction is necessary before using the SBP data sets in physical/conceptual models. However, there is no straightforward way of applying a correction scheme, especially for precipitation which includes a larger uncertainty range compared to temperature data sets. Additionally, bias correction is more convenient in case of having more reliable GBP data sets. Alternatively, data sets can be directly used in the modeling itself. This is possible in black-box models. Thus, secondly, the data sets are trained and tested with raw data, i.e., without applying a strict bias correction procedure.

Among the many types of modeling environments, artificial neural network models are becoming popular in hydrological applications because of their capacity to map nonlinear interactions. The MLP is the most often used form. It is a feedforward network with linked nodes (neurons) organized into three layers: an input layer, a hidden layer, and an output layer [57]. The layers of MLP are connected in parallel, and each layer has a variable number of nodes that receive information from the previous layer's nodes and pass it to the following layer's nodes. Because the information only flows via the front layers, it is referred to as feedforward.

Hidden layers are the layers that exist between the input vectors and the output layer. Each link is allocated a synaptic weight and bias to indicate the relative connection strength of two nodes at both ends in predicting the input–output relationship. The last or output layer consists of the network's projected values and hence represents model output. The

number of input and output nodes varies according to the problem type and data sets. Equation (1) yields the output of node *j*, $Q_j$.

$$Q_j = f\left(X{\cdot}W_j - b_j\right) \tag{1}$$

$$X = (X_1, \ldots, X_i, \ldots, X_n)$$
$$W_j = \left(W_{1j}, \ldots, W_{ij}, \ldots, W_{nj}\right)$$

where *X* is information from previous nodes, $W_{ij}$ represents the connection weight from the *i*th node in the preceding layer to this node, where $b_j$ is bias, f is the activation function.

Considering the purpose of the study, the lagged runoff values are not provided as input to the NNM model and input vectors only are generated by giving precipitation and temperature, as well as time-lagged data for both. The structure of the model is constructed using the cross-validation stopping criterion to avoid overtraining. The training model is used with at least 100 runs (the model gives comparable results, indicating the model's stability), and the results show average values. This eliminates initial weight dominance and enables for the random selection of cross-validation samples in each cycle.

*2.4. Assessment Criteria*

For the accuracy assessment, different goodness of fit criteria are defined. These are Percent Bias (P-BIAS, Equation (2)), Root Mean Square Error (RMSE, Equation (3)), and Pearson correlation coefficient (Pearson-R, Equation (4)). Two additional metrics, Nash–Sutcliffe Efficiency (NSE, Equation (5)) and Mean Absolute Error (MAE, Equation (6)), are used to test hydrological model performances. Pearson R and NSE values close to one are preferable by indicating the good consistency between model values and observation values. NSE is also sensitive to catching peak values in the hydrograph. The others are errors, and the minimum error is preferable. All performance metrics are calculated as:

$$P - BIAS = \frac{\sum_{t=1}^{n}\left(X_m^t - X_o^t\right)}{\sum_{t=1}^{n} X_o^t} \tag{2}$$

$$RMSE = \sqrt{\frac{\sum_{t=1}^{n}\left(X_m^t - X_o^t\right)^2}{n}} \tag{3}$$

$$Pearson\text{-}R = \frac{\sum_{t=1}^{n}\left(X_m^t - \overline{X}_m\right)\left(X_o^t - \overline{X}_o\right)}{\sqrt{\sum_{t=1}^{n}\left(X_m^t - \overline{X}_m\right)^2}\sqrt{\sum_{t=1}^{n}\left(X_o^t - \overline{X}_o\right)^2}} \tag{4}$$

$$NSE = 1 - \frac{\sum_{t=1}^{n}\left(X_o^t - X_m^t\right)^2}{\sum_{t=1}^{n}\left(X_o^t - \overline{X}_o\right)^2} \tag{5}$$

$$MAE = \frac{\sum_{t=1}^{n}\left|X_o^t - X_m^t\right|}{n} \tag{6}$$

where $X_m^t$ is product/estimate, $X_o^t$ is observation, $\overline{X}_m$ is average product/estimate, $\overline{X}_o$ is average observation, *n* is the number of data sets and *t* is the time index.

Contingency tables evaluate the consistency of the products by using two main indicators, i.e., Probability of Detection (POD) (Equation (7)) and False Alarm Rate (FAR) (Equation (8)). Four different possibilities are categorized: "*a*" (detected by both ground and satellite), "*b*" (detected by only satellite), "*c*" (detected by only ground), and "*d*" (no detection in both).

$$POD = \frac{a}{a + c} \tag{7}$$

$$FAR = \frac{b}{a + b} \tag{8}$$

Finally, P-BIAS, RMSE, Pearson-R, POD, and FAR are used for SBP comparisons, and Pearson-$R^2$, NSE, RMSE, and MAE are used for the assessment of hydrological modeling results.

## 3. Results

In this section, PERSIANN, PERSIANN-CDR, PERSIANN-CCS, and PERSIANN-PDIR-Now are renamed as PER, CDR, CCS, and PDIR to increase the readability of the text.

### 3.1. Product-Validation Experiments (PVEs)

3.1.1. Annual Product-Validation Experiments (PVEs)

The mean annual precipitation from 2003 to 2015 equals 488, 311, 862, 756, and 617 mm/year for GBP, PER, CCS, CDR, and PDIR, respectively. The initial assessment is demonstrated by "the mean biases" and "the annual variation of the mean biases" of the SBP data with respect to the GBP data ($\pm$ mm/year or $\pm$ mm/season) and the box-whisker diagrams of the SBP data along with the GBP data in Figure 5. According to the mean biases, the PER data underestimates in the WY and three seasons. Even though it is a pre-bias correction algorithm, the CDR data still contains positive biases. The highest mean biases are observed for the CCS data set in the WY and two seasons (accumulation and melting). For the WY, the lowest mean bias is detected from the PDIR data set (+124 mm/year), and the highest is detected from the CCS data set (+358 mm/year), while the others are equal to −169 mm/year and +257 mm/year for PER and CDR data sets, respectively. Three SBP (CDR, CCS, and PDIR) overestimates in the accumulation season, and only PER underestimates. The least bias is observed for the PER data set (−58 mm/season) in the same season, followed by the PDIR (101 mm/season), the CDR (+125 mm/season), and the CCS (+234 mm/season) data. For the melting season, the PDIR data shows the best performance (+22 mm/season) among all according to the GBP data comparison. The CCS data overestimates and is higher than the CDR data in this period. Finally, the least bias (almost zero) is detected from the PDIR data in the no-snow season, while the others still vary between −54 mm/season (PER) and +69 mm/season (CCS).

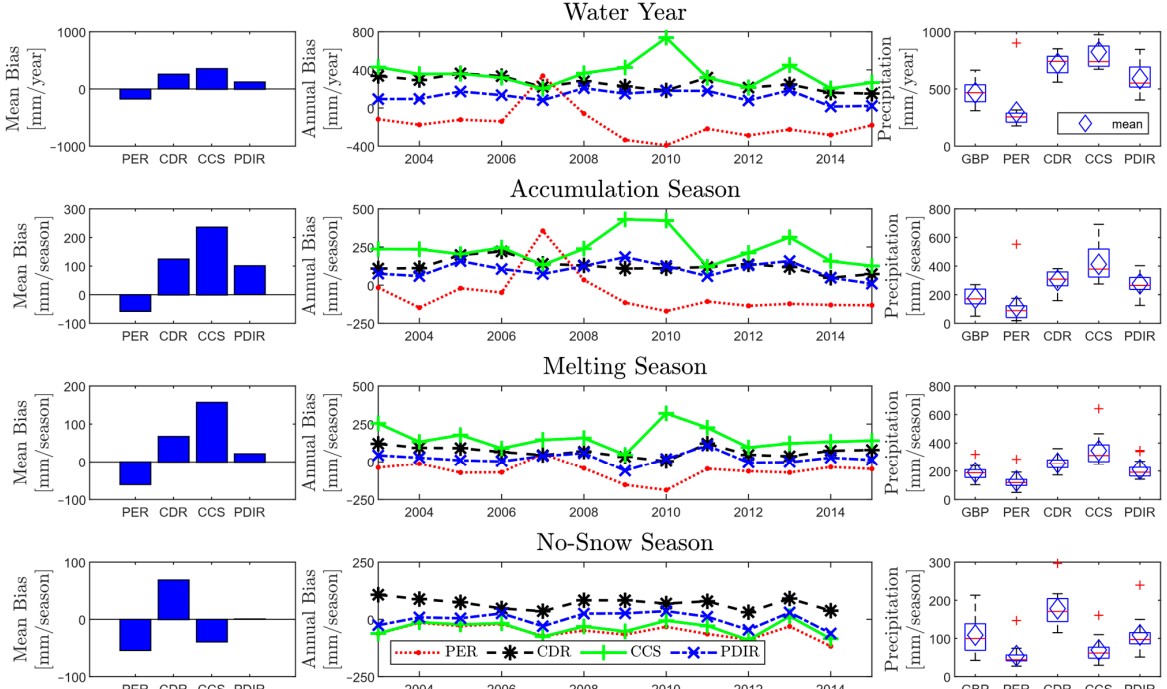

**Figure 5.** Annual PVE of SBP data set * GBP: Gauge-based Precipitation, PER: PERSIANN, CDR: PERSIANN-CDR, CCS: PERSIANN-CCS, PDIR: PERSIANN-PDIR-Now.

The trend of the annual biases of the SBPs in the WY is almost constant for each year except for some outlier cases, for example, in 2007 for PER and in 2009–2010 and 2013 for CCS. These outliers should be increasing the mean biases of the CCS data, but the variation of the biases over years in the CCS data is always higher than the other SBPs. The annual CDR and PDIR biases seem similar except for the WY, but the differences turn up in the seasonal scales. CCS shows a larger positive bias during the melting season. Less variation is detected during the no-snow season, and the highest bias is observed for the CDR data set.

The WY box-whisker diagrams demonstrate the higher CDR, CCS, and PDIR values (minimum, mean, median, maximum) and the diagram with whiskers does not neatly match with GBP spread. PDIR shows the larger spread, but mean and median values are still above the GBP data set. Annual SBP mean values of CCS and PDIR are higher than their corresponding median values in the WY. This might be due to including larger precipitation values in the SBP data. The highest spread is detected in the CCS data set for the accumulation and melting seasons. The narrower spread is observed for the melting season for all products. The noticeable overestimation and underestimation can be detected for CDR and PER in the no-snow season, respectively.

### 3.1.2. Monthly Product-Validation Experiments (PVEs)

Monthly PVE is conducted with two approaches: firstly, demonstrating the monthly box-whisker plots in Figure 6 (Supplementary Figure S1 presents a narrower y-axis range for July–September), and secondly, detecting the monthly relationships with scatter diagrams of monthly data sets in Figure 7. Monthly boxplots of all products (SBP and GBP) seem within a similar range for the beginning of the WY (October and November), only the PER data underestimates. However, the CCS data shows the highest mean and spread for continuing four months (between December and March). The highest difference between mean and median is observed for January and March for the CCS data. The larger spread is detected for the CDR data in April. Moreover, PDIR demonstrates a closer mean with GBP, but then the spread is narrow which might be associated with missing low and high precipitation events in January, Mar, April, and May. Minimum values are always high in the CDR, CCS, and PDIR data from December to May. The CDR data also gives the highest precipitation values for the late spring and summer months (June–September).

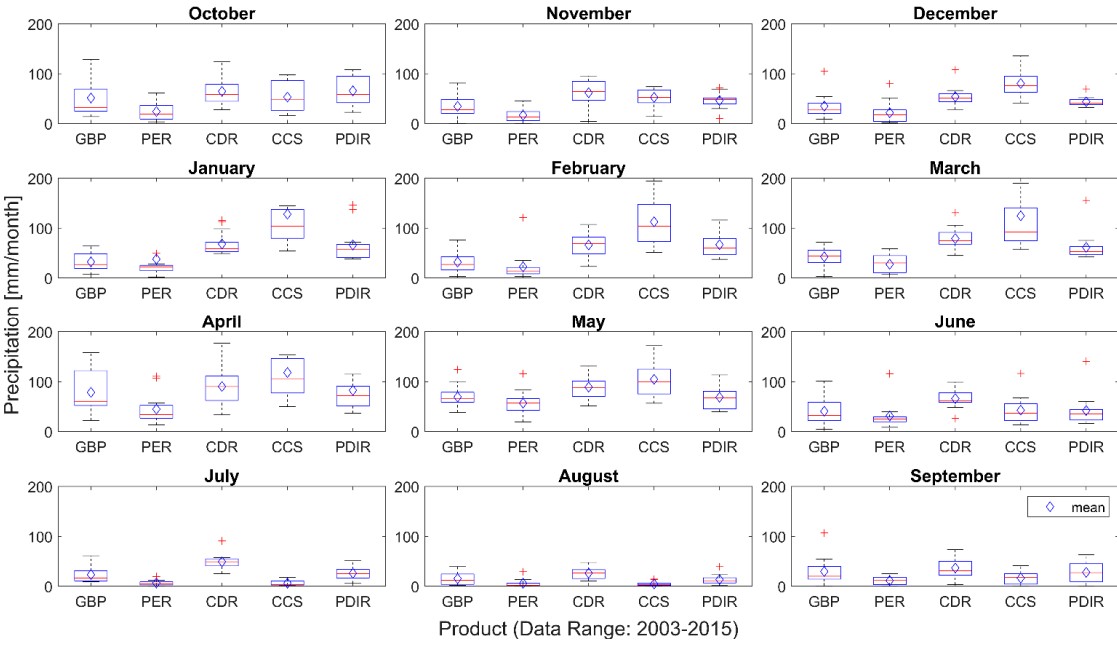

**Figure 6.** Monthly PVE of SBP data set with boxplots and whisker diagrams * GBP: Gauge-based Precipitation, PER: PERSIANN, CDR: PERSIANN-CDR, CCS: PERSIANN-CCS, PDIR: PERSIANN-PDIR-Now.

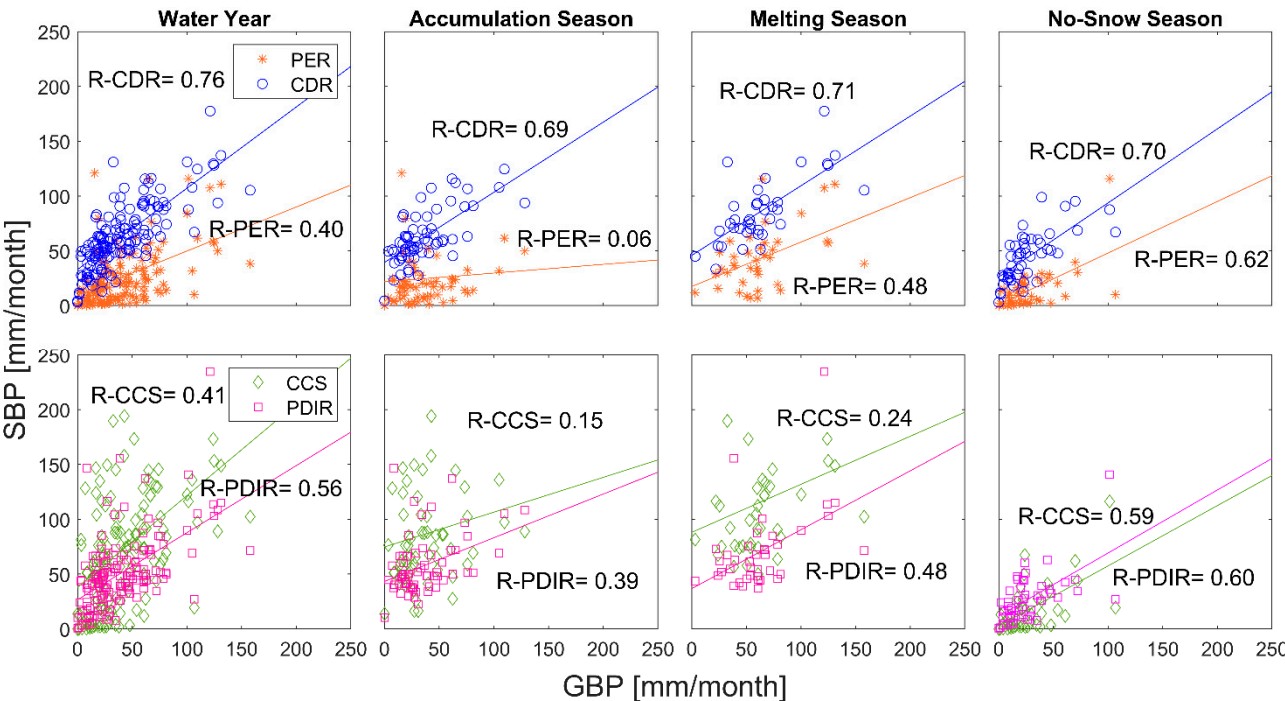

**Figure 7.** Monthly PVE of SBP data set with regression analysis (where *p* < 0.005 except for PER (*p* = 0.6526), CCS (*p* = 0.2377) during the accumulation season, and for CCS (*p* = 0.1403) during the melting season). * GBP: Gauge-based Precipitation, PER: PERSIANN, CDR: PERSIANN-CDR, CCS: PERSIANN-CCS, PDIR: PERSIANN-PDIR-Now.

The highest correlation between SBP with GBP is detected by the CDR data in three seasons and the WY. These relationships are significant (*p* < 0.005) except for PER (*p* = 0.6526) and CCS (*p* = 0.2377) during the accumulation season, and for CCS (*p* = 0.1403) during the melting season. These exceptions have also low R values (≤0.24). The CDR (R = 0.69) and PDIR (R = 0.39) data outperform the CCS and PER data which have almost zero relationships during the accumulation season. This is remarkable because the CDR data has the second largest mean bias compared to other SBP data. The PER data scatter reaches up to relatively better performance as much as the PDIR data (R = 0.48) for the melting season whereas CCS is the lowest (R = 0.24). The highest correlations are observed for the no-snow season for all SBP data sets; however, the maximum precipitation is around or less than 100 mm in the GBP. The highest performance in the WY is provided by the CDR data (R = 0.78), followed by the PDIR data (R = 0.58), the CCS data (R = 0.44), and the PER data (R = 0.42).

### 3.1.3. Daily Product-Validation Experiments (PVEs)

The daily PVEs are conducted with daily precipitation values against GBP and the statistics are calculated in Figure 8. The statistics are also conducted for various threshold precipitation values ($p \geq 0$ mm/day, $p \geq 1$ mm/day, $p \geq 2$ mm/day, and $p \geq 5$ mm/day) so that the effect of the precipitation intensity is analyzed. According to the results, similarly to the annual and monthly comparison, the CCS data gives the highest positive P-BIAS (overestimation) among all SBPs for the WY, the accumulation, and the melting seasons. Major biases of the CCS and CDR data occur during the accumulation season. Even the ratio reaches up to 150% for the CCS data. Increasing the threshold precipitation does not affect it. On the other hand, the PER data gives negative P-BIAS (underestimation) in the three seasons and the WY. Increasing the precipitation threshold decreases the P-BIAS of the PDIR data, but no drastic change is observed for the remaining SBP data sets.

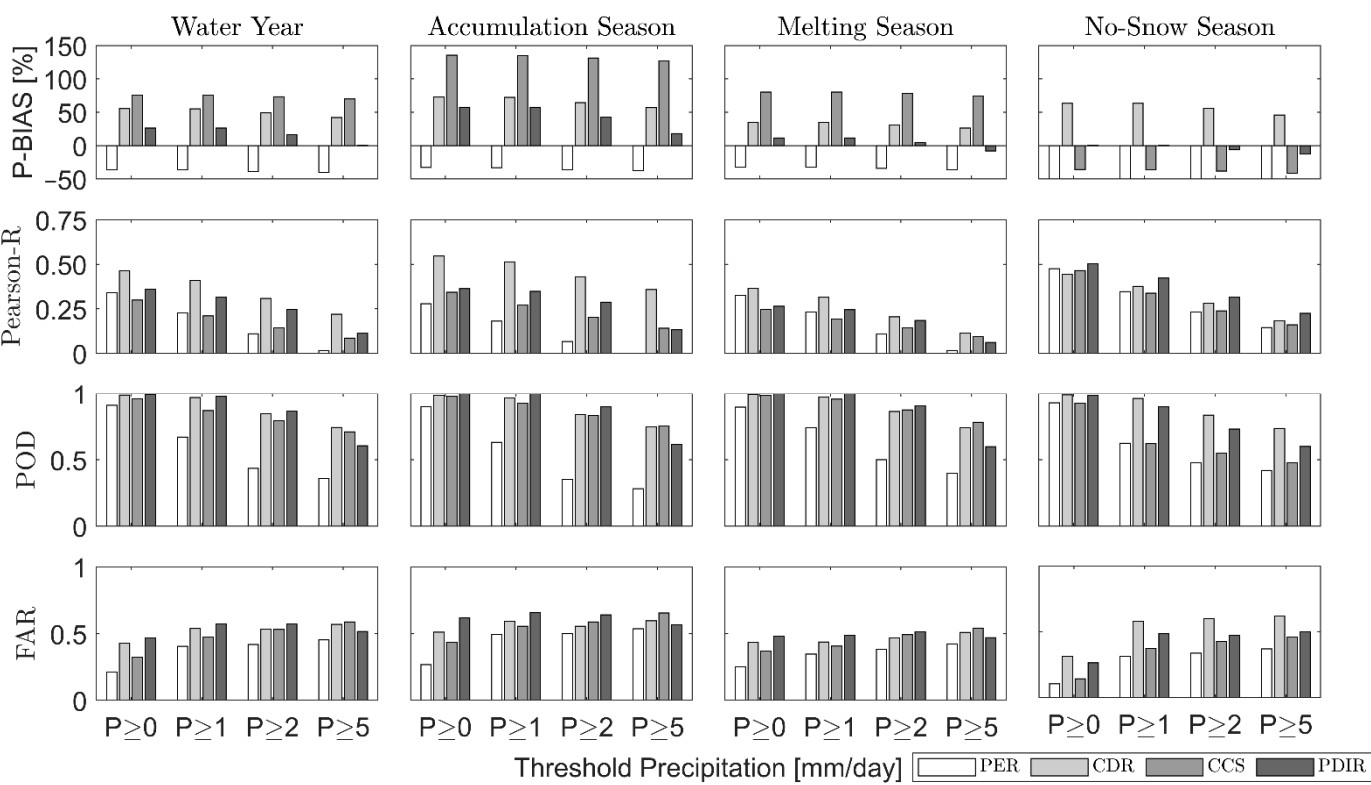

**Figure 8.** Daily PVE of SBP data set with performance criteria * GBP: Gauge-based Precipitation, PER: PERSIANN, CDR: PERSIANN-CDR, CCS: PERSIANN-CCS, PDIR: PERSIANN-PDIR-Now.

The correlations decrease in daily timer scale ($p \geq 0$ mm/day, CDR = 0.46, PDIR = 0.36, PER = 0.34, and PDIR = 0.30 for the WY). Even though the overestimation, the highest correlation (in terms of Pearson-R) is noticed for CDR data for all seasons except summer. This should be related to the pre-bias correction of the CDR data sets. It is also noteworthy that CDR outperforms for the accumulation period in terms of the R values. The correlation of the PER data sets shows the lowest performances and increasing the threshold precipitation critically decreases the R values. All SBP data sets except PER demonstrate higher POD values for different seasons. POD values are not decreasing drastically for the intense precipitation events; however, the FAR values are almost half which indicates the $\frac{1}{2}$ overestimation of the daily precipitation cases.

### 3.2. Hydro-Validation Experiments (HVEs)

In this section, hydro-validation of the different data sets (GBP and SBP) is employed with a neural network-based MLP rainfall-runoff model. MLP model results are compared with observed streamflow data. The first subsection describes the model performances of the MLP using observed GBP data; the second subsection describes the performances of different MLP models using different SBP data sets.

### 3.2.1. Daily Hydro-Validation Experiment (HVEs) using GBP Data

The MLP model outputs having observed precipitation (GBP) are demonstrated in Figure 9 (the GBP-based HVE). This experiment is also considered a benchmark model for SBP-based MLP models since GBP is assumed to have the lowest uncertainty. According to the results, the MLP model is capable of matching the hydrograph shape, snow melting process, and the low flow trend. The model is also sensitive to simulating high flow years (e.g., the 2007 water year) and low flow years (e.g., the 2008 water year) separately. Model performances are satisfactory for various metrics. For example, $R^2$ values are 0.91 and 0.81 and NSE values are 0.89 and 0.80 for training and testing periods. The model gives good accuracy especially up to 450 m³/s as shown in the scatter diagrams. Above this threshold,

the model underestimates the peak values but increasing the training performance to match rare peak events brings about the overtraining of the MLP model parameters. There is an extreme event that occurred during the 2004 water year. After analyzing the observation, this peak value is evaluated as a rain-on-snow case which has challenging melting processes, and the literature also mismatches these events even when applying different conceptual degree-day-based conceptual models [58]. Another general reason for the mismatching of the other peaks could be the lower representation of the precipitation in both GBP- and SBP-based models, which is similarly observed in the literature with other process-based models [34,37]. Additionally, the model can better represent the peaks and the low values depending on the selection of error criteria in the training part; however, this can cause a problem of overtraining and a loss of the ability to generalize, and may result in low performance, especially for the testing period.

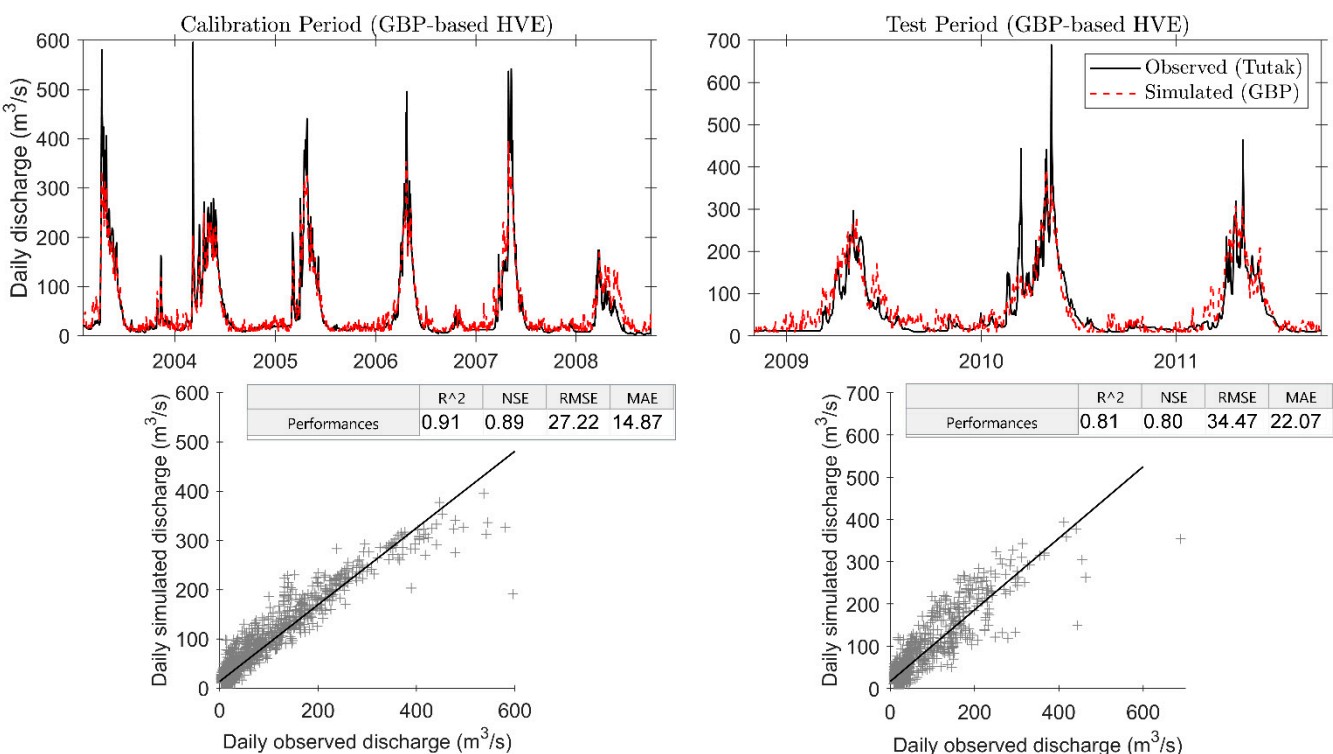

**Figure 9.** HVE results using Agri station GBP data.

### 3.2.2. Daily Hydro-Validation Experiment (HVEs) using SBP Data

The remaining SBP-based HVEs are accomplished with only the SBP data sets. The models are separately trained and tested using their own inputs/SBP data sets. This is because a data-driven model performs better with its own data sets. Figure 10 demonstrates the PER-based HVE with relatively high performance considering the underestimation of the PVE performance of PER data set. Model performances are satisfactory for the training period ($R^2$ = 0.87 and NSE = 0.85) but slightly decreased for the testing period (0.69 for both accuracy indicators). This might be attributed to the underestimation of peak events due to an increase in RMSE compared to GBP-based HVE. PERSIANN-CDR-based HVE results outperform PERSIANN-based HVE results with 0.88 and 0.79 NSE values for training and testing, respectively (Figure 11). The reduced RMSE in CDR-based HVE also verifies the enhanced simulation of the high flow values compared to PER-based HVE. CDR-based simulations result in a better mimic of the snowmelt runoff.

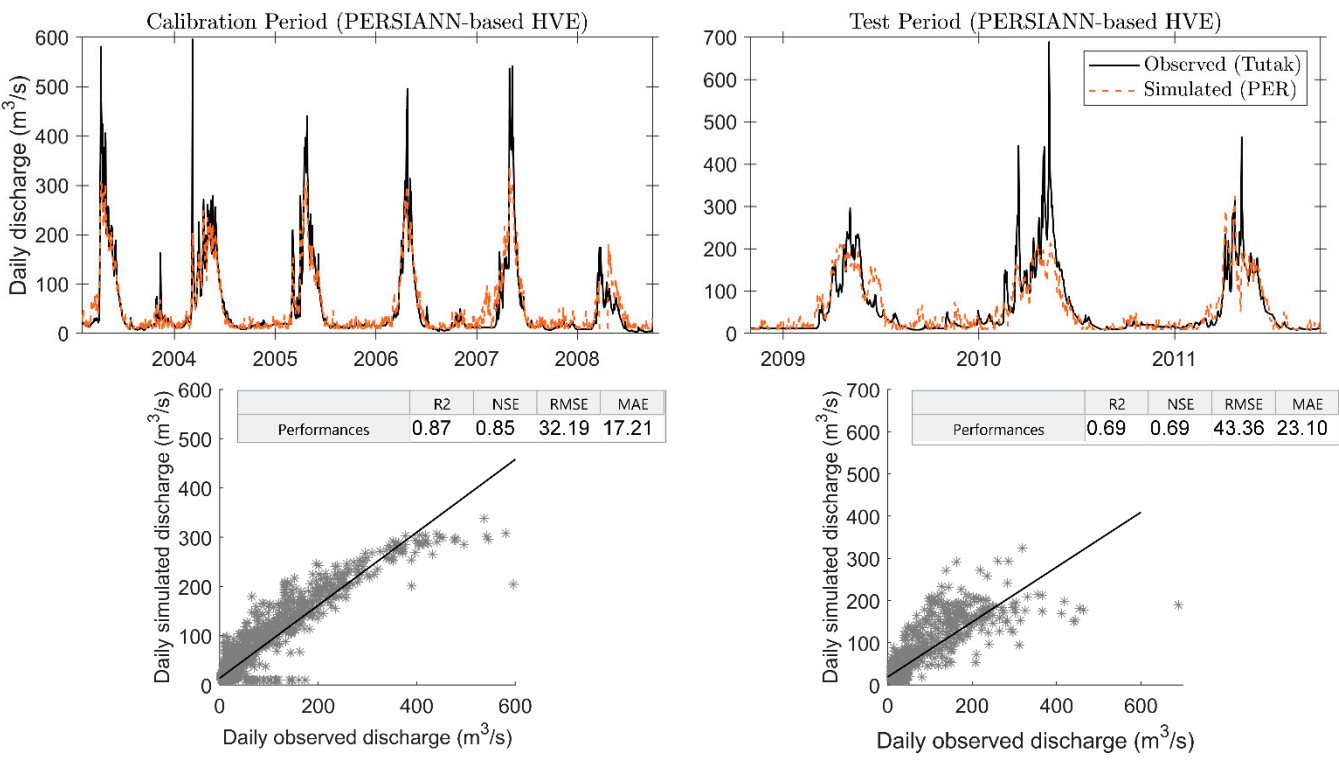

**Figure 10.** HVE results using PERSIANN SBP data.

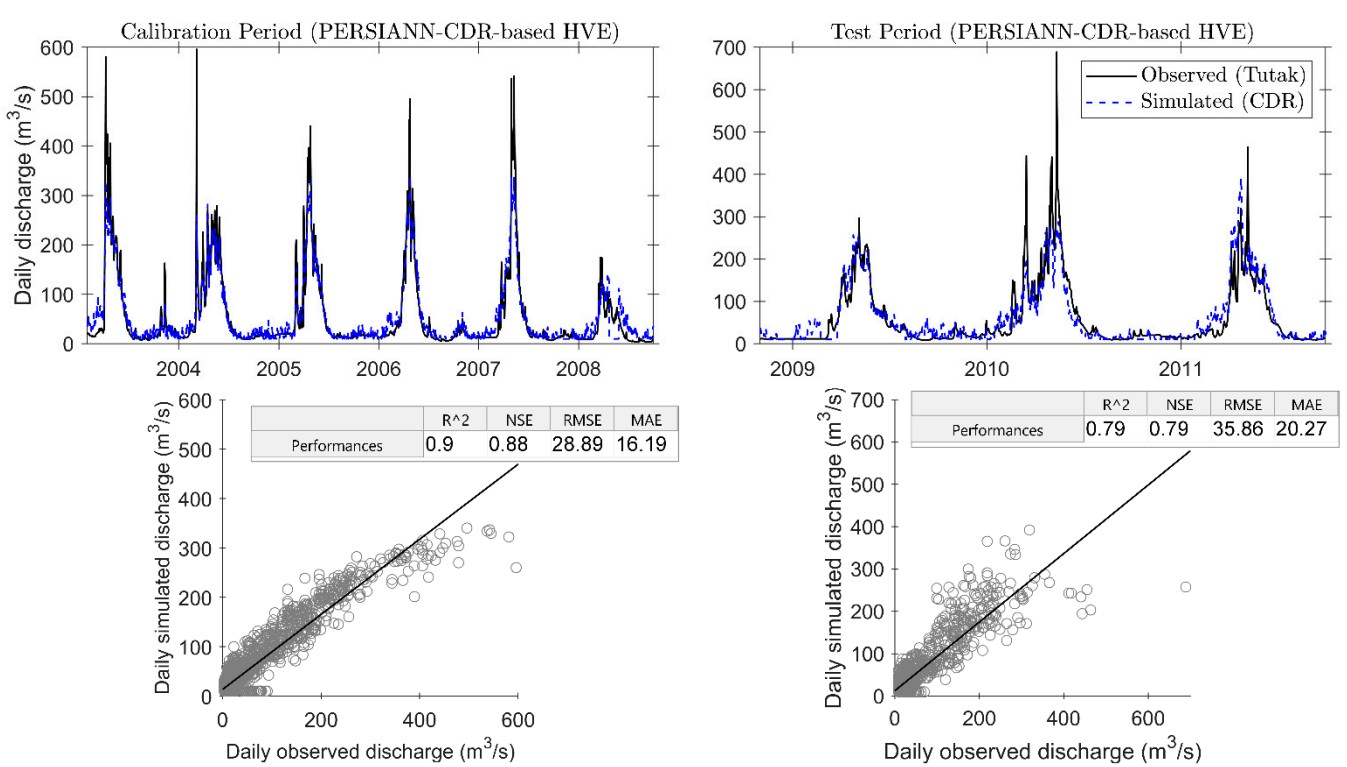

**Figure 11.** HVE results using PERSIANN-CDR SBP data.

High-resolution product (CCS and PDIR)-based HVEs (Figures 12 and 13) also demonstrate the capability of both SBP in streamflow simulation (with 0.85-0.80 NSE for the training period and 0.66–0.62 NSE for the testing period); however, they present relatively fewer NSE values compared to the low-resolution product (PER and CDR)-based HVEs.

The main discrepancies appear during the testing period, even though both (low- and high-resolution SBP-based) HVEs have close training performances. The main difference is detected due to overestimations in both low-flow and recession periods. For example, there is a high overestimation of the accumulation and recession of the hydrograph in 2009. This could be linked to larger FAR detected in the PVEs of the high-resolution products.

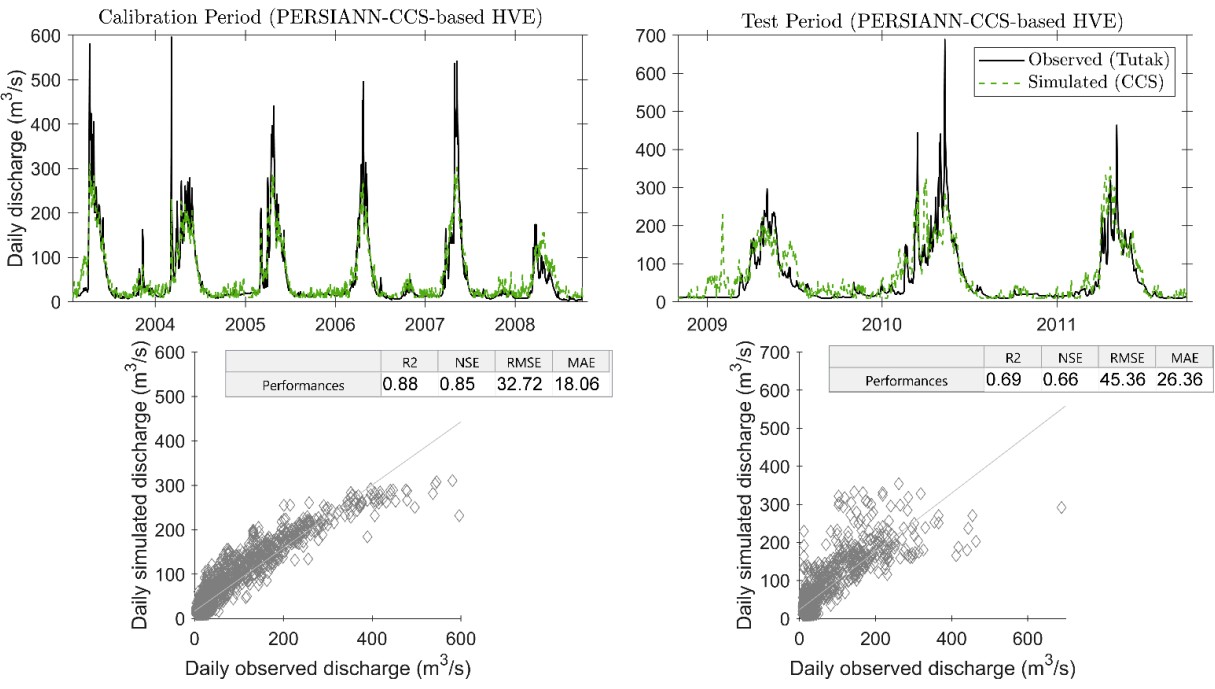

**Figure 12.** HVE results using PERSIANN-CCS SBP data.

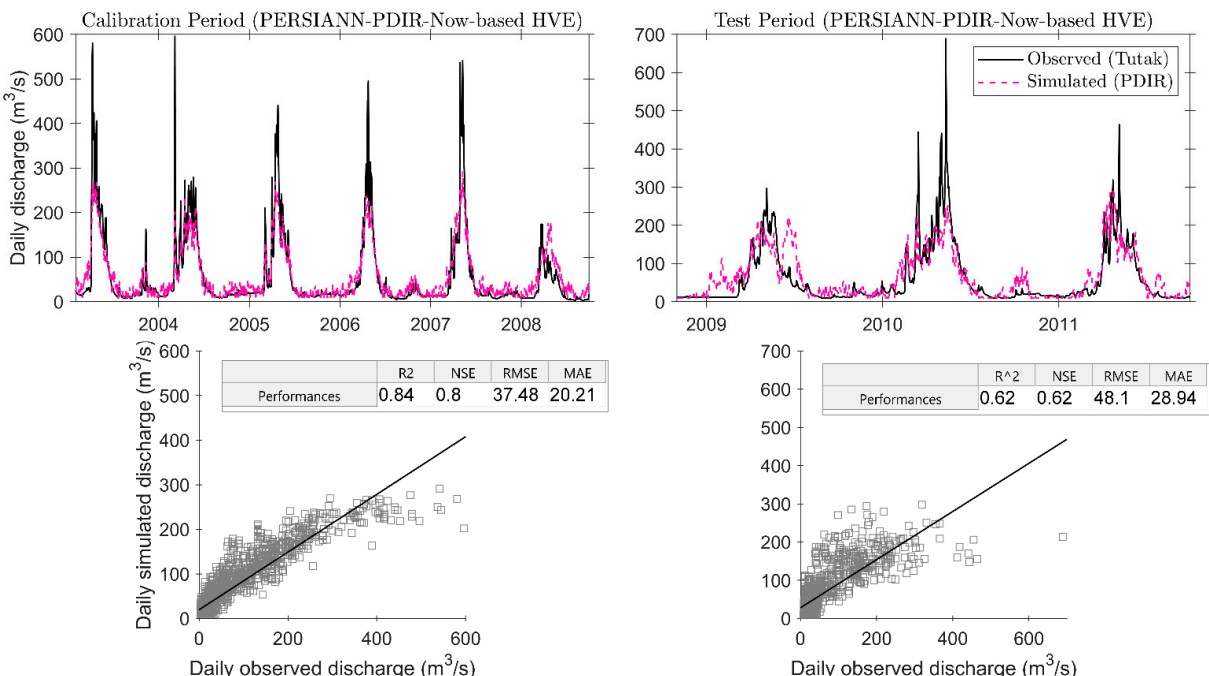

**Figure 13.** HVE results using PERSIANN-PDIR SBP data.

## 4. Discussion

### 4.1. Discussion of the Results

This section discusses the aforementioned PVE and HVE results. PVEs account for the validation of the SBP with GBP from 2003 to 2015, while HVEs account for analyzing the effects on the snowmelt runoff with observed streamflow data from 2003 to 2011. The variation of the annual biases does not change drastically among different years except for a few outliers (Figure 5). Table 3 summarizes major results obtained from the analyses of the annual biases. According to that, three SBPs (CDR, CCS, and PDIR) overestimate the precipitation in all seasons (except CCS in no-snow season), while PER underestimates in all seasons. The highest overestimation is detected for CCS data, especially in the accumulation and the melting seasons. This might be attributed to the cloud algorithm of the product which might not be capable of properly detecting snow precipitation. The lowest absolute is detected for the PDIR data. This could be evaluated as an enhancement since the PDIR system is an improved version of the CCS system. The CDR data is pre-bias corrected SBP using the GPCP data sets but still suffers from positive high biases, especially in the accumulation season. The CPCP data set uses gauge measurements, and the quality of the bias correction is also related to the gauge network in the selected region. Therefore, the performance of the bias-corrected CDR data could be different than the other regions [33], and thus, the biases can be attributed to insufficient data or bias correction algorithm for the mountainous regions. On the other hand, PDIR has the lowest biases for melting and no-snow seasons, but this could be related to some other factors such as missing daily high precipitation. Table 4 summarizes the monthly and daily Pearson-R correlation of each SBP with GBP in different seasons. Even though it has a low spatial resolution, the CDR data gives the highest monthly and daily correlation with the GBP data among all PERSIANN family SBPs except the no-snow season. Secondly, the PDIR data has the highest correlation in almost all seasons (the PER data is slightly better in the no-snow season). Lower similar relationships are detected for the daily and monthly PER and CCS data in the water year assessment; however, the PER and CCS data present a noticeable better daily correlation in the accumulation season. This could be related to some outlier events included in the daily data which do not affect the daily correlation but negatively affect the monthly ones. According to the analyses, it can be stated that the CDR and PDIR data show promising performances to use in meteorological or hydrological applications, while the CDR data has the highest capability to match fall precipitation that occurs as snow in mountainous regions like in this study. Additionally, the monthly correlations of the SBP against the GBP data are higher compared to daily relationships.

**Table 3.** The summary of the annual performances of PVEs (January 2003–May 2015).

| Data Type | Product * | Mean Annual Bias (mm/Year) | Mean Seasonal Bias in AS ** (mm/Season) | Mean Seasonal Bias in MS ** (mm/Season) | Mean Seasonal Bias in NS ** (mm/Season) |
|---|---|---|---|---|---|
| Low-resolution SBP | PER | −168 | −58 | −61 | −54 |
| | CDR | +256 | +125 | +68 | +69 |
| High-resolution SBP | CCS | +357 | +237 | +157 | −39 |
| | PDIR | +123 | +101 | +22 | 0 |

Note: * PER: PERSIANN, CDR: PERSIANN-CDR, CCS: PERSIANN-CCS, PDIR: PERSIANN-PDIR-Now. ** WY: Water Year, AS: Accumulation Season, MS: Melting Season, NS: No-snow Season.

The summary of HVEs is presented in Tables 5 and 6 for the training and testing of the daily MLP models, respectively. The highest performance (both $R^2$ and NSE) is obtained with the GBP-based HVE in both training (NSE = 0.89) and testing periods (NSE = 0.80). Secondly, the CDR-based HVE shows the highest performance in terms of $R^2$, NSE, and errors among SBP-based HVEs. The bias-corrected CDR, despite giving high biases, can provide successful modeling metrics in the hydrological application. However, the latency of the product is 3 months; thus, the CDR data could not be useful for daily operational purposes. The PER-based HVE shows superior performance compared to

the lower performance of PVE. Bearing in mind the possibility of real-time studies with short latency, the CCS- and PDIR-based HVEs still show good modeling skills (NSE above 0.80 and 0.62 for the training and the testing periods, respectively).

**Table 4.** The summary of the monthly and daily Pearson-R values of PVEs (January 2003–May 2015).

| Data Type | Product * | Water Year | | Accumulation Season | | Melting Season | | No-Snow Season | |
|---|---|---|---|---|---|---|---|---|---|
| | | Monthly | Daily | Monthly | Daily | Monthly | Daily | Monthly | Daily |
| Low-resolution SBP | PER | 0.40 | 0.34 | 0.06 | 0.28 | 0.48 | 0.28 | 0.62 | 0.47 |
| | CDR | 0.76 | 0.46 | 0.69 | 0.55 | 0.71 | 0.37 | 0.70 | 0.47 |
| High-resolution SBP | CCS | 0.41 | 0.30 | 0.15 | 0.34 | 0.24 | 0.25 | 0.59 | 0.47 |
| | PDIR | 0.56 | 0.36 | 0.39 | 0.36 | 0.48 | 0.26 | 0.60 | 0.50 |

Note: * PER: PERSIANN, CDR: PERSIANN-CDR, CCS: PERSIANN-CCS, PDIR: PERSIANN-PDIR-Now.

**Table 5.** The summary of the performances of HVEs (the calibration period, 2003–2008 WY period).

| Data Type | Utilized Product * in the Hydrological Model | $R^2$ | NSE | RMSE (m$^3$/s) | MAE (m$^3$/s) |
|---|---|---|---|---|---|
| In situ rain-gauge | GBP | 0.91 | 0.89 | 27.22 | 14.87 |
| Low-resolution SBP | PER | 0.87 | 0.85 | 32.19 | 17.21 |
| | CDR | 0.90 | 0.88 | 28.89 | 16.19 |
| High-resolution SBP | CCS | 0.88 | 0.85 | 32.72 | 18.06 |
| | PDIR | 0.84 | 0.80 | 37.48 | 20.21 |

Note: * GBP: Gauge-based Precipitation, PER: PERSIANN, CDR: PERSIANN-CDR, CCS: PERSIANN-CCS, PDIR: PERSIANN-PDIR-Now.

**Table 6.** The summary of the performances of HVEs (the testing period, 2009–2011 WY period).

| Data Type | Utilized Product * in the Hydrological Model | $R^2$ | NSE | RMSE (m$^3$/s) | MAE (m$^3$/s) |
|---|---|---|---|---|---|
| In situ rain-gauge | GBP | 0.81 | 0.80 | 34.47 | 22.07 |
| Low-resolution SBP | PER | 0.69 | 0.69 | 43.36 | 23.10 |
| | CDR | 0.79 | 0.79 | 35.86 | 20.27 |
| High-resolution SBP | CCS | 0.69 | 0.66 | 45.36 | 26.36 |
| | PDIR | 0.62 | 0.62 | 48.10 | 28.94 |

Note: * GBP: Gauge-based Precipitation, PER: PERSIANN, CDR: PERSIANN-CDR, CCS: PERSIANN-CCS, PDIR: PERSIANN-PDIR-Now.

Finally, MLP-based streamflow simulations from HVEs are analyzed in three periods in terms of errors and mean runoff values ($\overline{Q}_m$) in Table 7. The error analysis presents similar performances within the seasons so that the lowest errors are obtained for CDR and PER-based models apart from the GBP-based model. The mean snowmelt runoff obtained from PER and PDIR-based HVEs show lower than the mean observed runoff (previously given in Table 2). This might be due to them that they cannot properly capture the precipitation, especially during the snow melting period.

Snow physics is complex, and it is difficult to represent it with either a black box or a conceptual model (even with a physically based model due to limited data). Bearing in mind the atmospheric and topographical conditions, installing online automatic stations, collecting reliable data, and having a rich observation network is very challenging in snow-dominated high-altitude regions. Some other reasons related to the quality of GBP data are explained in detail in the "Limits and the Constraints of the Study Area Section". Moreover, the snow accumulation and melting processes are also governed by the temperature as well. The quality of the temperature data is effective in the melting process. From the

modeling perspective, improving the model development by fuzzy/hybrid systems would be of concern for future studies in better peak runoff forecasting.

**Table 7.** Seasonal assessment of the modeled flows.

| Data Type | Utilized Product * in the Hydrological Model | Accumulation Season | | | Melting Season | | | No-Snow Season | | |
|---|---|---|---|---|---|---|---|---|---|---|
| | | $\overline{Q}_m$ (m³/s) | RMSE (m³/s) | MAE (m³/s) | $\overline{Q}_m$ (m³/s) | RMSE (m³/s) | MAE (m³/s) | $\overline{Q}_m$ (m³/s) | RMSE (m³/s) | MAE (m³/s) |
| In situ rain-gauge | GBP | 25.7 | 17.7 | 12.0 | 163.1 | 57.7 | 43.2 | 40.5 | 24.1 | 25.7 |
| Low-resolution SBP | PER | 22.0 | 16.1 | 10.4 | 137.4 | 77.9 | 52.5 | 38.4 | 24.4 | 22.0 |
| | CDR | 23.2 | 16.9 | 11.7 | 150.8 | 64.0 | 44.4 | 35.3 | 18.0 | 23.2 |
| High-resolution SBP | CCS | 31.4 | 29.1 | 17.2 | 157.4 | 75.8 | 52.8 | 45.2 | 26 | 31.4 |
| | PDIR | 31.1 | 26.1 | 17.1 | 138.4 | 76.8 | 54.2 | 49.6 | 38.6 | 31.1 |

Note: * PER: PERSIANN, CDR: PERSIANN-CDR, CCS: PERSIANN-CCS, PDIR: PERSIANN-PDIR-Now.

Finally, Hafizi and Şorman [13] demonstrate that daily PVEs of PERSIANN and PERSIANN-CCS using daily scatter diagram and linear correlation have similar or better performances with some other SBPs such as TMPA 3B42T V7 and TMPA 3B42RT V7. However, when SBPs are employed in TUW hydrological model to simulate streamflows, they also find out that PERSIANN and PERSIANN-CCS show one of the lowest NSE among 13 SBP data sets while TMPA 3B42T V7 and TMPA 3B42RT V7 still give quite high performances. This might be related to the greatest p-biases (under- and overestimation) detected for PERSIANN and PERSIANN-CCS in PVE compared to GBP in their study. Similar p-biases (under- and overestimations) are also obtained in my study for PERSIANN and PERSIANN-CCS, respectively. Conversely, my case demonstrates that HVE performances of PERSIANN and PERSIANN-CCS are quite higher compared to what they found in their study. This shows the promising capability of the neural network-based MLP modeling which can provide better performances even if SBP has larger p-biases.

### 4.2. Limits and the Constraints of the Study Area

SBP data used in this study is freely accessible to all users. The applicant can download for a selected region/watershed. The data might require bias correction before using in a meteorological/hydrological model application. However, this study prefers to calibrate and validate each model parameter using raw data and aims to assess their practical usage of them in a hydrological monitoring system. Therefore, it can be stated that integrating SBP with the ANN model provides to generate daily streamflows for remotely located, snow-dominated, and poorly gauged regions.

Although precipitation and temperature are fundamental variables for the melting process, these variables can only be observed at point stations. It is always challenging to represent the areal distribution of these variables, especially in the higher zones of mountainous regions. Most stations installed by governmental organizations in Turkey are located near city centers and flat lands. The main assumption of this study is that there is only one station located in the lower parts of the terrain. The station is assumed to be an average of a relatively large area. This can affect the PVE of the SBP and cause higher p-bias values of the products than actual. This is observed especially for low-resolution PERSIANN-CDR and high-resolution PERSIANN-CCS data. On the other hand, considering the lapse rate higher precipitation is expected in the upper part of the terrain. This assumption is assessed with scatter diagrams and Pearson-R values and PERSIANN-CDR demonstrates high relationships against high p-biases, which is not the case for PERSIANN-CSS. From this, it can be assumed that the performance of PERSIANN-CDR can be high in the upper zones against PERSIANN-CCS data. Considering this limitation, the study performs both PVE and HVE. HVE is the main point of this study since there is only one GBP in the basin. Additionally, even though one station is assumed to be the average of the catchment, GBP still gives the highest performance for HVE. Thus, this indicates the reliable capability of gauge representation. On the other hand, underestimation of peak

discharges might be associated with a low representation of intense precipitation. This can be improved in further studies using new gauge data sets, in various different snow dominated regions for both meteorological and hydrological applications.

Despite its bias-corrected daily product, PERSIANN-CDR has a slightly coarse resolution, making it difficult to use in small catchments with high topographical complexity. This study, however, demonstrated that coarse resolution could be used successfully, particularly in snow-dominated mountainous regions with medium and/or large basins.

## 5. Conclusions

Data scarcity is a major problem that hinders the development of hydrological applications which we need more than before due to the global challenges [59]. Precipitation is the most vital input for hydrological monitoring and modeling studies. Readily available free SBP data sets could be one valuable alternative for the modelers and operators. Therefore, this study compares four different SBPs from the PERSIANN family (PERSIANN, PERSIANN-CDR, PERSIANN-CCS, and PERSIANN-PDIR-Now) in a data-sparse snow-dominated mountainous region of the upper part of the Euphrates River Basin. The analyses are conducted in various aspects such as considering different time scales (annual, monthly, and daily) against ground-based measurement, and a hydro-validation of the products in a neural network-based MLP model with the observed runoff. Each product has different advantages, so the main conclusions drawn from the application can be listed as follows together with a future outlook:

- The PERSIANN-CDR data gives the best monthly and daily correlation with GBP within the water year season and is followed by the PERSIANN-PDIR-Now, PERSIANN-CCS, and PERSIANN data. All products reveal the highest bias errors in the snow accumulation season for monthly and daily assessment. Despite high correlations and pre-bias correction algorithms, PERSIANN-CDR data still holds large biases which indicates that its utilization in meteorological and hydrological studies should be carefully done. This might be due to the low density of rain gauges used in the bias correction in the data-scarce mountainous regions. On the other hand, PERSIANN-CDR system provides historical records which go back to 1983. This study shows that PERSIANN-CDR-based HVE by a neural network can facilitate filling the streamflow gaps in ungauged/poorly gauged mountainous basins.

- The lower performance of PERSIANN-CCS might be attributed to the insufficient cloud algorithm, so the snow precipitation is extremely overestimated, especially in the accumulation season. Instead of PERSIANN-CCS data, the dam operators can utilize the enhanced form of it (PERSIANN-PDIR-Now data) which demonstrates the lowest mean bias and reasonable monthly correlation. Additionally, PERSIANN-PDIR-Now is a near-real-time and high-resolution product and holds the potential to be used in the real-time operation of the dams.

- In all SBP data sets, monthly correlations are higher than daily correlations, which indicates the daily mismatches. This is only vice versa in the accumulation period of the PERSIANN and PERSIANN-CCS data which might be related to outlier years. POD values do not significantly decrease during heavy precipitation events; nevertheless, FAR values are almost half, indicating the overestimation of daily precipitation instances.

- Even though high positive (PERSIANN-CDR, PERSIANN-CCS, and PERSIANN-PDIR-Now) and negative biases (PERSIANN) were obtained compared to ground precipitation data sets, in general, hydro-validation experiments show the promising capability of the utilization of SBPs to account for the daily neural network-based MLP rainfall-runoff model.

- The inter-comparison of the basins could reveal the potential insights of the SBPs in different regions for future studies. Additionally, the study can be extended through different climate regions, time resolutions, and elevation zones across Turkey.

**Funding:** This study was partly supported by the Eskisehir Technical University Scientific Research Project (Project No: 21GAP087).

**Supplementary Materials:** The following supporting information can be downloaded at: https://www.mdpi.com/article/10.3390/w14172758/s1, Figure S1: Monthly PVE of SBP data set with boxplots and whisker diagrams for July–August. * GBP: Gauge-based Precipitation, PER: PERSIANN, CDR: PERSIANN-CDR, CCS: PERSIANN-CCS, PDIR: PERSIANN-PDIR-Now.

**Data Availability Statement:** Publicly available free datasets are analyzed in this study.

**Acknowledgments:** Special thanks are extended to the Governmental Organization State Hydraulics Works (DSI) and State Meteorological Directorate (MGM).

**Conflicts of Interest:** The authors declare no conflict of interest.

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
