# Peer review of "Product- and Hydro-Validation of Satellite-Based Precipitation Data Sets for a Poorly Gauged Snow-Fed Basin in Turkey"

_water, doi:10.3390/w14172758_

Round 1

Reviewer 1 Report

This paper is well written and designed. The introduction properly underlines the novelties of the presented work and contextualizes it in the present scientific literature. The results are well commented. I would recommend its publication on WATER.

Author Response

Thank you for your time in reviewing my paper and your positive comments.

Reviewer 2 Report

General Comments:

This manuscript focused on validation of satellite-based precipitation (SBP) products for the upper Euphrates Basin in Turkey. The research topic is interesting, and I think it may contribute to ungauged basin on choosing the satellite-based precipitation products to some extent. However, I feel that the manuscript needs further revisions and clarifications, before consideration for publication. Moreover, similar study for mountainous basin of Turkey (one of the tributaries of same Euphrates River Basin) has been already conducted by Hafizi and Sorman (2022), and this study followed the same approach. Therefore, it is not clear that what are the new contributions of this paper, including novel parts and new findings. In addition, the topographical condition of this study area is steep terrain ranging from 1500 to 3500 m and also catchment size is 5,910 sq km. However, author evaluated performance of SBP using only precipitation data of one station that is located in lower part, and performance of SBP might be completely different in high mountain areas. Therefore, author should also discuss on possible variations on assessment results due to data limitation, including how assessment can be further improved. Please also see specific comments below.

Specific Comments:

1.      Line 29, “In these basins,.” in which basins? mountainous basins? snowmelt contributes to the runoff only in the snow fed mountainous basins, but not in all mountainous basins in the globe.

2.      Lines 65-66, please indicate the region.

3.      Line 69, please define full form of VIC.

4.      Lines 76-78, please elaborate more on how SBP can provide better performance than gauge precipitation in general with more evidences, not only by limited study with limited gauge data.

5.      Lines, 99-105, since similar study has been already conducted for mountainous areas of Euphrates Basin in Turkey, please clarify why this study is crucial and what is novel part of this study compared to previous study.

6.      Lines 106-108, there are numerous studies, which focused both precipitation and hydro validation of satellite-based precipitation and also with daily time steps. Therefore, authors should do further intensive reviews of those paper and emphasize on importance of this study. For instance, see following papers

https://doi.org/10.3390/hydrology8040154

https://doi.org/10.5194/hess-2016-473

https://doi.org/10.1016/j.ejrh.2022.101109

https://doi.org/10.1080/02626667.2016.1154149

https://doi.org/10.1016/j.ejrh.2020.100768

https://doi.org/10.3390/rs14133127

https://doi.org/10.21203/rs.3.rs-1712655/v1

https://doi.org/10.3390/rs13020221

7.      Lines 144-146, I think total number of homepage visitors should not be the reason for choosing the SBP products. Authors should emphasize more on reliability or accuracy/quality of the product rather than number of visitors/register users.

8.      Lines 147-149, please clarify why all five SBP products were not utilized in this study, what are the reasons for choosing only four types SBP for the study?

9.      Table 1, Time delay of PERSIANN product is 2-day, generally, the near-real time products should be available within a few hours of the observation.  Also, please add time delay for PERSIANN-CCS and PERSIANN-PDIR-Now in the table. Is it 1hr or less than 1hr?

10.  Line 172, what is the catchment size of Euphrates River Basin?

11.  Line 178, E21A022 ?

12.  Lines 187-188, it is a big assumption for steep terrain with areas of 5,910 sq km. The performance of SBP products might be completely different in higher altitude areas than in lower part of study area.

13.  In Fig. 2, please show the location of study area at least in Turkey or in western Asia. Please also show the boundary of Euphrates River Basin (if possible) and location of Tutak gauge station in the map.

14.  In Fig.3, please add source for observed data.

15.  Line 235, what do you mean by OBP?

16.  Lines 282-290, these descriptions are completely repeated and could be deleted.

17.  Lines 349-3598, please clarify that how those relationships between GBP and SBP were statistically significant.

18.  Fig. 7, I suggest to use same corresponding color as in scatter points data for each trend line.

19.  Sections 3.2.1-3.2.1, even though the model performance are in satisfactory based on metrics measures, all model including GBP-based HVE could not reproduce any of the high peaks of discharge. So, I recommend to discuss some possible reasons.

20.  Lines 514-515, I recommend to discuss not only capturing performance of the precipitation by the satellite products, but also what factors were considered in this study to reflect snow melting in model fitting, how considered factor plays role in snow melting process, and how the issues of underestimation of snowmelt runoff can be improved in further study.

21.  Conclusions, the repeated descriptions from background and method could be deleted to make it more concise.

Reviewer 3 Report

This study evaluated four PERSIANN products in a data-sparse snow-dominated mountainous region of the upper part of the Euphrates River Basin. Moreover, the author conducted the hydro-validation as well. Although the study is interesting, I found some points that need to be improved.

I think the most significant weak point of this study is that there is only one rain gauge in a 5910 km2 basin area. This could show a lack of representation of the results.

PERSIANN-CCS and PERSIANN-PDIR-Now have a high resolution of 0.04°. However, the author used 0.25°for the evaluation, which usually can not meet the requirement of practical demands.

The y-axis scale should be improved in Fig. 5 (too broad and not easy to get info. From them) and in Fig. 6.

Round 2

Reviewer 2 Report

Author substantially revised the manuscript addressing the comments and the revised version of manuscript is much improved. However, I still have some minor comments that need to be addressed.

1.      Lines 56, 68, and 73, please also add country of Heihe River basin, Illinois River basin, and Cerrado biome. Please also check in other places.

2.      Line 81, SWAT is already defined in lines 71-72, so, it is not necessary to define it again at multiple places. Please also check in other places and other acronyms/abbreviations throughout the manuscript.

3.      Lines 85-96, please define full form of acronyms/abbreviations at its first use. Also, I have noticed that some acronyms/abbreviations are defined at multiple places (For example, SWAT as mentioned in comment 2, MLP in lines 115, 162 273). Please check these throughout the manuscript.

4.      Lines 176-178, reasons for choosing four SBP products out of five are not given in the text.

5.      Line 217,  at 1632 m --> at an altitude of 1632 m ??

6.      In Legend of Fig. 1, Tutak DEM (m) should be placed below legend for Euphrates river basin

7.      Lines 376-377, please define PPER, PCCS.

8.      Figure 6, for better comparison among the graphs, the scale of the Y-axis of all the graphs in Figure 6 should be unified, I think “0 to 200” in all figures would be best.  

9.      Figure 7, it is not clear that which trend line corresponding to which scatter data plot. I suggested to use same color in each trend line that is used in the corresponding scatter data plot (for example, red color trend line for red colored scatter data, blue color trend line for blue colored scatter data). Also, please remove the y-axis labels in the bottom figure for accumulation season.

10.  In Figures 9-13, it would be best if author could plot the scatter plots in square box with same scale for both x- and y-axis.

Reviewer 3 Report

I appreciate the authors' efforts to revise the manuscript. Now it looks good for publication.

Author Response

(The authors gave the same response as above.)
